# Gaze Fixation and Visual Searching Behaviors during an Immersive Virtual Reality Social Skills Training Experience for Children and Youth with Autism Spectrum Disorder: A Pilot Study

**DOI:** 10.3390/brainsci12111568

**Published:** 2022-11-18

**Authors:** Thomas David Elkin, Yunxi Zhang, Jennifer C. Reneker

**Affiliations:** 1Department of Psychiatry & Human Behavior, School of Medicine, University of Mississippi Medical Center, 2500 North State Street, Jackson, MS 39216, USA; 2Department of Pediatrics, School of Medicine, University of Mississippi Medical Center, 2500 North State Street, Jackson, MS 39216, USA; 3Department of Data Science, John D. Bower School of Population Health, University of Mississippi Medical Center, 2500 North State Street, Jackson, MS 39216, USA; 4Department of Population Health Sciences, John D. Bower School of Population Health, University of Mississippi Medical Center, 2500 North State Street, Jackson, MS 39216, USA; 5Department of Physical Therapy, John D. Bower School of Population Health, University of Mississippi Medical Center, 2500 North State Street, Jackson, MS 39216, USA

**Keywords:** autism spectrum disorder, virtual reality, social skills, eye tracking

## Abstract

Children and youth with Autism Spectrum Disorder (ASD) display difficulties recognizing and interacting with behavioral expressions of emotion, a deficit that makes social interaction problematic. Social skills training is foundational to the treatment of ASD, yet this intervention is costly, time-consuming, lacks objectivity, and is difficult to deliver in real-world settings. This pilot project investigated the use of an immersive virtual reality (IVR) headset to simulate real-world social interactions for children/youth with ASD. The primary objective was to describe gaze fixation and visual search behaviors during the simulated activity. Ten participants were enrolled and completed one social-skills training session in the IVR. The results demonstrate differential patterns between participants with mild, moderate, and severe ASD in the location and duration of gaze fixation as well as the patterns of visual searching. Although the results are preliminary, these differences may shed light on phenotypes within the continuum of ASD. Additionally, there may be value in quantifying gaze and visual search behaviors as an objective metric of interventional effectiveness for social-skills training therapy.

## 1. Introduction

The number of children and youth diagnosed with Autism Spectrum Disorder (ASD) has been on the rise over the past 20 years, with current estimates showing that 1 in 44 children meet criteria for this disorder in the United States [1]. Children and youth diagnosed with ASD demonstrate, “persistent deficits in social communication and social interaction”, and “restricted, repetitive patterns of behavior” [2]. Despite the rising prevalence of ASD, methodological differences in diagnosis and symptoms differ within and between healthcare professionals, and the symptoms themselves have changed over the years [2]. Healthcare professionals are united, however, on their desire to improve the quality of life for these individuals. Attempts to treat symptoms of ASD have focused on normalizing social communication and interaction, both verbal and non-verbal, and decreasing the repetitive behaviors.

Researchers and clinicians have sought to address deficits in nonverbal communicative behaviors in children with ASD, including intensive clinical interventions aimed at improving behavioral difficulties manifested socially [3]. These behavioral difficulties range from not readily recognizing social space, indifference to others, persistent and repetitive behaviors and interests, and self-stimulation behavior (e.g., hand-flapping). Another component of nonverbal communicative behaviors is eye contact. Often, clinicians will instruct youth with ASD to “make eye contact when speaking”, but the efficacy of this is problematic due to the difficulty with obtaining a measure of eye contact in a clinical context. Rather, the measurements are restricted to estimations (e.g., “a lot”, “not much”), which are neither objective nor reproducible.

Immersive virtual reality (IVR) has recently emerged into healthcare and research as a novel tool, providing a new modality with which to interact. Commercially available IVR headsets can be used to provide a computer-generated, 3-dimensional, controlled, and reproducible environment for visually based tasks. It has been suggested that IVR enables the assessment of complex cognitive, behavioral, and motoric functions in a superior manner to traditional measures [4]. Additionally, researchers have discovered that IVR outperforms computer-based video technology in investigations of visual perception [5]. With the recent addition of eye trackers in the IVR headsets, unique research is being pursued. There is rising interest and research being undertaken to explore IVR as an interventional tool in children and adults with ASD [6,7,8].

In recent years, several applications and platforms have been developed to promote social skills training, emotion recognition, and language development for people with autism [6]. In general, the findings of these interventions have been positive, showing the transfer of gains made in the virtual environment to real-world interactions and scenarios [6,9]. Within the research on VR as an intervention for social and communication development in children with ASD, we have found no studies that describe visual behaviors of participants during the VR intervention. The objective of this study was to describe gaze fixation characteristics and visual searching behaviors within a sample of children/youth with ASD. First, we wanted to describe these visual outcomes in respect to a virtual avatar, as well as other locations in a virtual room. Secondarily, we wanted to describe the visual outcomes based on ASD severity. Describing these characteristics and behaviors are important as gaze and visual searching could be valuable targets for social skills training and provide objective metrics of change in response to intervention. 

## 2. Materials and Methods

### 2.1. Study Design

This is a prospective case-series, conducted as a pilot project.

### 2.2. Setting

This study was conducted in one calendar year, with enrollment of eligible participants between September 2020 and September 2021. Participants were recruited from the patient case of the developmental and behavioral outpatient treatment center at the University of Mississippi Medical Center (UMMC). Each participant completed one experience utilizing the virtual reality social skills training application. 

### 2.3. Participant Eligibility

Children and youth (ages 6–17 years old) who had a diagnosis of ASD and were seen for therapy in the Center for Advancement of Youth (CAY) at UMMC were eligible for enrollment. The principal investigator called the caregiver of an eligible participant and asked if they would be willing to have their child take part in the study. Caregivers who agreed over the phone were scheduled a time to come into the clinic, where written permission and assent were obtained. 

The diagnostic criteria for ASD from the Diagnostic and Statistical Manual of Mental Disorders, fifth edition (DSM-5)-were utilized [2]. The sample was further classified as mild, moderate, or severe based on the criteria listed in DSM-5. Each participant had been seen clinically in the CAY Center by a qualified healthcare provider trained to diagnose and treat children and youth with ASD, including developmental/behavioral pediatricians, child psychologists, and psychiatric mental health nurse practitioners. The classification of ASD severity was determined clinically, using the Autism Diagnostic Observation Schedule (ADOS) [10], administered in our clinic by trained professionals, after a thorough interview by both a psychologist and a developmental-behavioral pediatrician. The cut-off scores from the ADOS were used to determine ASD severity.

A total of 105 eligible participants were approached about this study; 77 did not respond to our initial invitation to participate; 17 expressed interest in participating but then could not be scheduled due to scheduling issues. Eleven participants were enrolled into the study.

### 2.4. Participant Orientation

Prior to entering the virtual experience, the participant chose the physical characteristics of their avatar from options that ranged in height, sex, and skin color. The purpose of this was to allow the participant to feel most “at home” with the avatar, rather than to have one chosen for them. The participant was allowed to wear the VR headset for a brief amount of time to ensure comfort with the system. All of the participants expressed an understanding of VR systems and an interest in them, which speaks to the relative ubiquity of VR in our culture. Despite this, several caregivers expressed concern that their child might not be comfortable having a VR headset on their head due to sensory issues, but only one eligible participant was unable to keep the VR headset on to participate.

### 2.5. Simulator Type

The HTC Vive Pro Eye VR headset provided the virtual environment and was tethered to an Alienware, Area 51 laptop computer, which ran the software application (VR SAFE Version I) with Tobii eye tracking integration through Steam and collected the sensor-based data. VR SAFE, was developed for this project. The software system produced a virtual school room to provide the social context for the training activities. The eye tracker measured the eye gaze of the participant within the virtual environment. It tracked how often the participant looked into the eyes of the avatar, and for how long, as well as where else the participant looked in the virtual environment. 

### 2.6. Virtual Environment

Once the choice of the participant’s avatar was made, the participant entered a virtual classroom with a whiteboard on a wall, a desk that had a globe on it, and an avatar.

### 2.7. Virtual Test Scenario

The avatar asked a series of five [5] questions of the participant, starting with forced choice answers and ending with open-ended questions. The questions ranged from “*what is your name?*” and “*what grade are you in?*” to “*what class are you going to?*” “*what do you like about your classes?*” and “*what else do you like to do?*” The responses given by the participant were entered into the dataset but were not part of the overall study, since we were interested in eye gaze at this time. The entire procedure took about 10 min to complete.

### 2.8. Variables and Data Sources

Participant data (age, sex, race, and ASD severity) were collected. All VR data were collected while the participant was in the virtual reality environment (see Figure 1). To capture the full detail of the child’s experience in the VR SAFE simulation, two types of data were collected that complement one another. The first is the data defining the environment, which describes the initial state of the simulation, including the 3D gaze-intersection boundaries for individual objects in the simulation. These objects may not correspond to recognizable entities in the simulation (e.g., a person); instead, they may correspond to an individual part of that entity (e.g., an eye, an arm, the back of a chair, the legs of a chair, etc.). The second is the simulation data, which acted as a log of the interactions that occurred throughout the participant’s interaction with the application. It referenced a given environment file version which served as a basis for any changes to the environment that may occur because of the child’s actions or general evolution of the scene during the conversation containing the child’s gaze and any changes in the environment that might impact the interpretation of the child’s gaze. 

The gaze positions were defined using Tobii software XR SDK v1.8.0 development kit (SDK), which is an analytical application for eye tracking data collected from Tobii pro (Danderyd, Sweden). Data were collected at a fixed update interval of each 0.01 s. A ray cast from the gaze position in the direction of the gaze vector was performed until one of the tracked objects was hit, which is considered “looked at”. If no tracked object was hit, then we consider the gaze on a null area. A gaze duration was calculated by subtracting the starting timestamp from the ending timestamp. 

### 2.9. Quantitative Variables

From the VR session, we recorded the child’s gaze on the blackboard, chairs, right controller, desks, globe, light switch, poster, teacher’s chair, teacher’s desk, wall colliders, floor collider, null area, and avatar. We further classified the gaze on the avatar’s left and right eyes, mouth, other face area, left and right hands. The number of times and the number of seconds of gaze were recorded. The primary outcome of this study is the number and duration of eye contacts between the participant and the avatar in the VR. The secondary outcomes of this study are the number and duration of child’s gaze on mouth and face area of the avatar.

### 2.10. Statistical Methods

Mean and standard deviation (SD) were used to summarize the number and the duration of gaze on the left and right eyes, mouth, and face of the avatar. We calculated them for all participants and by ASD level. Transition probabilities from avatar’s face area to an environmental object or other areas of the avatar were also calculated for all participants and by participant ASD level.

## 3. Results

### 3.1. Participants

Among the 11 participants enrolled, 10 completed the VR session, and one participant who had a severe level of ASD was not able to keep the VR headset on to complete the session. Among the ten VR session completers, seven participants (70%) had mild ASD, two participants (20%) had a moderate level of ASD, and one participant (10%) had a severe level of ASD. The average age of all enrolled participants was 12.53 (SD 2.24) years old.

### 3.2. Outcome Data: Gaze Fixations

The average number of gazes on left and right eyes were 14.5 (SD 26.3) and 21.8 (SD 36.2), which correspond to average durations of 2.8 (SD 2.8) seconds and 3.1 (SD 4.0) seconds, respectively (Table 1). Among all face areas, participants gazed on the mouth 95.4 (SD 166.1) times with the longest duration of 6.7 (SD 11.4) seconds (Table 2). The average number of gazes on other face areas, neither eyes nor mouth, was 106.2 (SD 168.8), with an average duration of 4.6 (9.3) seconds. Further results of the number of gaze fixations and duration of gaze on object and avatar (non-face area) are reported in Appendix A, respectively.

In the case of the participant with severe ASD, there was no gaze on the face. Participants with moderate ASD had more gaze and longer durations on all face areas compared to those with mild ASD. The average number of gazes on the left eye area of participants with moderate ASD was 35.5 (SD 46.0), with an average duration of 2.9 (SD 2.7) seconds, while the average number of gazes on the left eye area of participants with mild ASD was 4.0 (SD 2.6), with an average duration of 2.6 (SD 3.2). The participants with moderate ASD had 76.0 gazes on the right eye area with 3.3 (SD 4.1) seconds of duration, and participants with mild ASD had 3.7 (SD 2.1) gazes with 1.7 (SD 1.7) seconds of duration. Participants with moderate ASD had 230.5 (SD 320.3) gazes with an average duration of 7.1 (SD 12.3) seconds on the mouth area and 224.5 (SD 311.8) gazes with an average duration of 5.2 (SD 10.4) seconds on the other face area. Participants with mild ASD had 41.4 (SD 54.4) gazes with an average duration of 5.6 (SD 9.0) seconds on the mouth area and 47.0 (SD 33.2) gazes with an average duration of 3.2 (SD 5.9) seconds on the other face area.

### 3.3. Outcome Data: Visual Searching Behavior

It was most likely that after gazing at the left and right eye areas, participants would look at the other face area, with an average probability of 0.557 (SD 0.357) and 0.412 (0.476), respectively (Table 3). After gazing at the left eye area, if transiting to a non-avatar object, moderate level ASD participants may focus only on the floor collider with a probability of 0.022 (SD 0.031), whereas mild level ASD participants may look at multiple objects, including the chair (probability 0.111 (SD 0.192)) and wall collider (probability 0.095 (SD 0.165)), but not the floor collider. After gazing at the right eye area, mild level ASD participants did not look at environmental objects, while a moderate level ASD participants gazed on the floor collider with a probability of 0.053.

Participants with mild and moderate levels of ASD may look at different objects or areas after gazing at the mouth area. Mild level ASD participants may look around and are mostly likely to transit gaze to the wall collider 2 (probability 0.274 (SD 0.362)), poster (probability 0.250 (SD 0.500)), and other face area (probability 0.248 (SD 0.207)), while moderate level ASD participants may transit gaze to floor collider with a probability of 0.694 (SD 0.079) or other face area (probability 0.298 (SD 0.068)).

After gazing at the other face area, if participants still looked at avatar’s face, mild level ASD participants were likely to gaze at the mouth area (probability 0.239 (SD 0.167)), and moderate level ASD participants were likely to gaze at the left eye are (probability 0.433 (SD 0.448)). If transiting to an environmental object, participants with a mild level of ASD gazed at several objects, including desk 1 (probability 0.005 (SD 0.008)), wall collider 1 (probability 0.042 (SD 0.072)), wall collider 2 (probability 0.380 (SD 0.375)), and floor collider (probability 0.257 (SD 0.433)), whereas participants with a moderate level of ASD transited only to the floor collider (probability 0.196 (SD 0.276)).

## 4. Discussion

The current study reports on the findings from a social skills training exercise in IVR, with specific attention to the gaze and visual searching behaviors of children with ASD.

### 4.1. Gaze Fixations

In this pilot study, we found that children/youth with moderate ASD had between 4 and 19 times as many gaze fixations with various face areas on the virtual avatar than those with mild ASD. Interestingly, the average time of each gaze fixation was not different between groups. Participants also spent more time with gaze fixation on the avatar’s mouth than on the avatar’s eyes. This finding aligns with a recent VR based job interview activity for adults with autism, which demonstrated that participants with ASD consistently demonstrated less eye contact and more gaze directed at the mouth when in the role of speaker or listener than neurotypical individuals [11].

In general, clinicians recognize that children/youth with more severe ASD tend to look into the eyes of other humans less frequently than those with mild ASD. Our results were surprising and indicated that the participants with moderate ASD looked into the avatar’s eyes more frequently than those with mild ASD. Recent research reflects that eye contact during conversation is specifically lacking during conversation activities [12], as with the avatar interaction in this VR simulation. It may be that youth with moderate ASD are more comfortable looking at the VR avatar because they see this as a videogame and, therefore, less threatening and more socially comfortable than a real-world setting. 

### 4.2. Visual Searching Behaviors

Additional findings describe patterns of eye movement behavior, related to the face of the avatar (to whom they are speaking) as well as other places and objects within the virtual classroom. Here, we found that individuals with moderate ASD demonstrated a generally greater propensity to transition their gaze to the floor after making eye contact with various locations on the avatar’s face than the participants with mild ASD.

We suggest that the VR environment may not present a similar experience to individuals with ASD. It may be that the VR environment was so novel that participants with mild ASD were enthralled with the VR environment and visually searched the room to “take it all in”, rather than interacting with the avatar as we had hoped. It was, however, confirmatory, that participants with moderate ASD displayed a tendency to shift their gaze to the floor as a preferential location, a behavior not observed in the participants with mild ASD. 

### 4.3. Related Work and Clinical Implications

Other researchers are investigating the use of VR for social interaction and training social skills in children and youth with ASD [6]. Most of these studies are primarily interested in the transfer of the training activities from VR to real-world applications. Results have provided promising evidence in support of VR based social interventions as effective modalities to improve real-world social skills and interactions [13,14,15]. Some reports of improvement in social interaction have been based on subjective teacher and parent reports [14]. Other research in this area has collected a child’s response to social interactions in IVR on a sliding scale or via multiple choice [13]. More objectively measured results suggest that IVR may be useful for identifying and categorizing social information processing (SIP) challenges and overall SIP ability for individual children with ASD [13].

Related research has demonstrated that eye gaze patterns and visual attention in an IVR simulated environment may be a useful biomarker for ASD diagnosis [16]. Outside of the IVR environment, gaze preferences between social scenes and non-social scenes [17], as well as pupillary responses during visual scanning activities have also been suggested as potentially useful in diagnostics for ASD [18]. We believe our method of tracking eye gaze and visual searching behavior during the IVR intervention adds to these findings and could augment the subjective perception of improvement and provide more objective measures and quantification of change in response to training. Further, the use of eye tracking in the IVR environment could potentially assist in characterizing SIP and changes over time in response to intervention. These are both areas we plan to explore in subsequent studies. 

### 4.4. Limitations

The results presented here are based on a small pilot sample of children and youth with ASD and may not be generalizable to the larger population with ASD. Additionally, this study utilized only one measurement of gaze and visual searching behavior. These propensities may rapidly evolve with subsequent exposures in IVR, which needs future exploration. Second, our novel IVR environment included a rather cartoonish version of a human avatar. One programming element was that the mouth on the avatar was the only moving part of the face or body after the avatar approached the participant, which may account for the number of gazes on the mouth as opposed to the eyes. 

## 5. Conclusions

Social skills training is an important component of therapeutic intervention for children and youth with ASD. IVR provides the possibility of longer and more frequent therapeutic training sessions, and related evidence has demonstrated promising results of effectiveness. The results of this study, in terms of the measurement of gaze fixation and visual searching behaviors during a simulated training session, reveal interesting patterns in individuals with mild and moderate ASD. The use of this objective measurement of gaze behavior during social skills training may be useful to these types of interventional applications and provide enhanced identification of individual social interaction propensities. 

## Figures and Tables

**Figure 1 brainsci-12-01568-f001:**
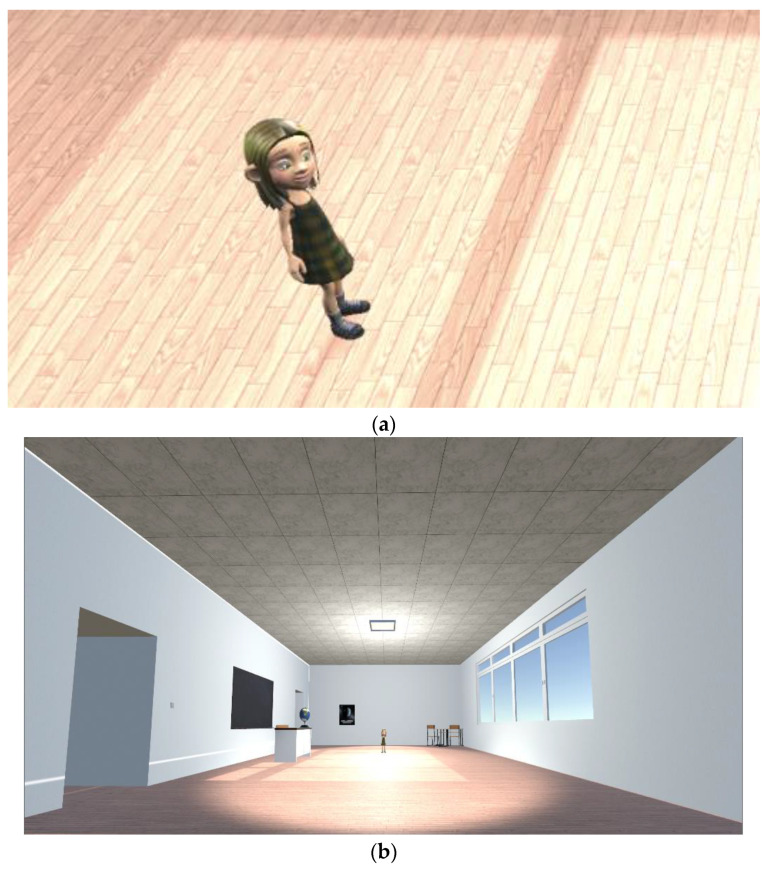
Snapshots of the VR session environment and avatar. (**a**,**b**) display a close-up and a broader view of the environment and avatar in the VR session, respectively.

**Table 1 brainsci-12-01568-t001:** The number of gaze fixations on the face of avatar, mean (SD).

Gaze Area	All Participants (N = 10)	Mild (n = 7)	Moderate (n = 2)	Severe (n = 1)
Left eye area	14.5 (26.3)	3.7 (3.1)	35.5 (46.0)	- (-)
Right eye area	21.8 (36.2)	4.0 (2.8)	76.0 (-)	- (-)
Mouth area	95.4 (166.1)	42.5 (62.8)	230.5 (320.3)	- (-)
Other face area	106.2 (168.8)	52.3 (38.5)	224.5 (311.8)	- (-)

**Table 2 brainsci-12-01568-t002:** The duration of each gaze episode on the face of avatar, mean (SD), seconds.

Gaze Area	All Participants (N = 10)	Mild (n = 7)	Moderate (n = 2)	Severe (n = 1)
Left eye area	2.8 (2.8)	2.6 (3.2)	2.9 (2.7)	- (-)
Right eye area	3.1 (4.0)	1.7 (1.7)	3.3 (4.1)	- (-)
Mouth area	6.7 (11.4)	5.6 (9.0)	7.1 (12.3)	- (-)
Other face area	4.6 (9.3)	3.2 (5.9)	5.2 (10.4)	- (-)

**Table 3 brainsci-12-01568-t003:** Gaze transition probability from the face of avatar, mean (SD).

Transition from	Transition to	All Participants (N = 10)	Mild (n = 7)	Moderate (n = 2)	Severe (n = 1)
Left eye area	Avatar				
	Right eye area	0.012 (0.030)	0.000 (0.000)	0.037 (0.052)	- (-)
	Mouth area	0.231 (0.385)	0.381 (0.541)	0.022 (0.031)	- (-)
	Other face area	0.557 (0.357)	0.302 (0.287)	0.919 (0.114)	- (-)
	Object				
	Chair 1	0.056 (0.136)	0.111 (0.192)	0.000 (0.000)	- (-)
	Wall collider 2	0.048 (0.117)	0.095 (0.165)	0.000 (0.000)	- (-)
	Floor collider	0.041 (0.080)	0.000 (0.000)	0.022 (0.031)	- (-)
	null	0.056 (0.136)	0.111 (0.192)	0.000 (0.000)	- (-)
Right eye area	Avatar				
	Left eye area	0.225 (0.233)	0.250 (0.354)	0.066 (-)	- (-)
	Mouth area	0.183 (0.222)	0.333 (0.236)	0.066 (-)	- (-)
	Other face area	0.412 (0.476)	0.417 (0.589)	0.816 (-)	- (-)
	Object				
	Wall collider 2	0.083 (0.167)	0.000 (0.000)	0.000 (-)	- (-)
	Floor collider	0.096 (0.160)	0.000 (0.000)	0.053 (-)	- (-)
Mouth area	Avatar				
	Left eye area	0.003 (0.005)	0.002 (0.004)	0.007 (0.009)	- (-)
	Right eye area	0.009 (0.016)	0.009 (0.185)	0.001 (0.002)	- (-)
	Other face area	0.250 (0.156)	0.248 (0.207)	0.298 (0.068)	- (-)
	Object				
	Poster	0.143 (0.378)	0.250 (0.500)	0.000 (0.000)	- (-)
	Wall collider 1	0.048 (0.126)	0.083 (0.167)	0.000 (0.000)	- (-)
	Wall collider 2	0.157 (0.295)	0.274 (0.362)	0.000 (0.000)	- (-)
	Floor collider	0.385 (0.363)	0.124 (0.215)	0.694 (0.079)	- (-)
	null	0.005 (0.014)	0.009 (0.185)	0.000 (0.000)	- (-)
Other face area	Avatar				
	Left eye area	0.174 (0.286)	0.028 (0.037)	0.433 (0.448)	- (-)
	Right eye area	0.051 (0.058)	0.032 (0.045)	0.073 (0.103)	- (-)
	Mouth area	0.226 (0.159)	0.239 (0.167)	0.173 (0.245)	- (-)
	Object				
	Desk 1	0.002 (0.006)	0.005 (0.008)	0.000 (0.000)	- (-)
	Wall collider 1	0.021 (0.051)	0.042 (0.072)	0.000 (0.000)	- (-)
	Wall collider 2	0.206 (0.306)	0.380 (0.375)	0.000 (0.000)	- (-)
	Floor collider	0.269 (0.315)	0.257 (0.433)	0.196 (0.276)	- (-)
	null	0.051 (0.099)	0.018 (0.021)	0.125 (0.177)	- (-)

## Data Availability

Data used in this manuscript may be obtained through written data use agreements with the University of Mississippi Medical Center with request via email to the corresponding author.

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
