# Peer review of "Gaze Fixation and Visual Searching Behaviors during an Immersive Virtual Reality Social Skills Training Experience for Children and Youth with Autism Spectrum Disorder: A Pilot Study"

_brainsci, 2022, doi:10.3390/brainsci12111568_

Round 1

Reviewer 1 Report

The authors conducted a descriptive cohort study of objective eye gaze measurements to assess the influence of the use of virtual reality intervention (VRI) with youth with ASD in changing visual gaze to eyes, face, hands, of an avatar, and objects within a virtual environment. The study is interesting and addresses an emerging area of interest within eye tracking applications to better understand ASD. The study design and results are weak, however, in findings and implications for connecting those findings to existing literature using eye tracking to understand visual attention in ASD. I recommend the authors consider bolstering their background literature review and discussion section by citing more references specifically focused on visual attention and eye tracking to social vs. non social stimuli, in order to draw more robust comparisons to a)what we already know about ASD and visual attention based on eye gaze to pictures and video, and b) what should be expected in outcome based on this knowledge, and what was a unique outcome because of the use of the avatar and VRI. Some relevant papers on the topic, including use of avatar and video technology is by Martineau et al., 2011 https://pubmed.ncbi.nlm.nih.gov/21679969/ and Shaffer et al., 2017 https://pubmed.ncbi.nlm.nih.gov/27878742/ .

Specific edits and consideration for improving the paper:

Introduction:

1.     line 25, update the current prevalence of ASD and cite, 1 in 44

2.     line 60, consider a stronger statement re: the rationale and purpose for conducting the study, needs to be more than it is “of interest”, how will this knowledge inform treatment, how will it help us better understand the nuances of visual attention in ASD?

Materials and Methods:

1.     Line 75, DSM “5”, not a roman numeral, correct “DSM V” to “DSM 5”, and spell out first use fully.

2.     What were exclusion criteria, if any? Meds vs. no meds? History of seizures were these excluded or included?

3.     A little more detail should be provided re: how levels of severity are defined for ASD under the current criteria "Level 1, Level 2, Level 3" is typically used, and you should include the DSM5 definition for each, and how this was determined---ADOS scores? Clinical judgment?  This applies later in the analysis section as well---provide cut-off scores if used, in this determination as well as “verbal” vs. “non verbal”?

4.     Move the total participant numbers cited in the results up to the methods section.

Results:

1.      Line 163, “Among 163 the ten VR session completers, seven participants (70%) had mild ASD, two participants 164 (20%) had a moderate level of ASD, and one participant (10%) had a severe level of ASD. 165 The average age of all enrolled participants was 12.53 (SD 2.24) years old.” How was this justified and defined based on endophenotype?

Discussion:

1.      It is well established that individuals with ASD pay more attention to the eyes than the mouth----this is not surprising----the authors need to tie in their findings more closely with existing literature----what is new that this approach adds to our current knowledge? Cite the relevant literature and connect in terms of where the use of video and visually based interventions are headed.

Author Response

Reviewer 1: The authors conducted a descriptive cohort study of objective eye gaze measurements to assess the influence of the use of virtual reality intervention (VRI) with youth with ASD in changing visual gaze to eyes, face, hands, of an avatar, and objects within a virtual environment. The study is interesting and addresses an emerging area of interest within eye tracking applications to better understand ASD. The study design and results are weak, however, in findings and implications for connecting those findings to existing literature using eye tracking to understand visual attention in ASD. I recommend the authors consider bolstering their background literature review and discussion section by citing more references specifically focused on visual attention and eye tracking to social vs. non social stimuli, in order to draw more robust comparisons to a)what we already know about ASD and visual attention based on eye gaze to pictures and video, and b) what should be expected in outcome based on this knowledge, and what was a unique outcome because of the use of the avatar and VRI. Some relevant papers on the topic, including use of avatar and video technology is by Martineau et al., 2011 https://pubmed.ncbi.nlm.nih.gov/21679969/ and Shaffer et al., 2017 https://pubmed.ncbi.nlm.nih.gov/27878742/ . – thank you for these suggestions and the additional references. Modifications were made throughout, especially in the discussion to bring these points forward.

Specific edits and consideration for improving the paper:

Introduction:

  1. line 25, update the current prevalence of ASD and cite, 1 in 44 – this was corrected.
  2. line 60, consider a stronger statement re: the rationale and purpose for conducting the study, needs to be more than it is “of interest”, how will this knowledge inform treatment, how will it help us better understand the nuances of visual attention in ASD? – this sentence was modified to be more assertive and active. It now reads, “The objective of this study was to describe gaze fixation characteristics and visual searching behaviors within a sample of children/youth with ASD. First, we wanted to describe these visual outcomes in respect to a virtual avatar, as well as other locations in a virtual room. Secondarily, we wanted to describe the visual outcomes based on ASD severity. De-scribing these characteristics and behaviors are important as gaze and visual searching could be valuable targets for social skills training and provide objective metrics of change in response to intervention.”

Materials and Methods:

  1. Line 75, DSM “5”, not a roman numeral, correct “DSM V” to “DSM 5”, and spell out first use fully. – this was corrected as suggested. Thank you
  2. What were exclusion criteria, if any? Meds vs. no meds? History of seizures were these excluded or included? – there were no exclusion criteria. This was added to the methods in line 85-86.
  3. A little more detail should be provided re: how levels of severity are defined for ASD under the current criteria "Level 1, Level 2, Level 3" is typically used, and you should include the DSM5 definition for each, and how this was determined---ADOS scores? Clinical judgment?  This applies later in the analysis section as well---provide cut-off scores if used, in this determination as well as “verbal” vs. “non verbal”? Thank you for pointing out a need to provide additional details. For easier interpretation of the levels, we assigned the qualitative descriptor of mild, moderate, and severe ASD. The severity of ASD was determined clinically, using the ADOS, administered in our clinic by trained professionals, after a thorough interview by both a psychologist and a developmental-behavioral pediatrician. We used the cut-off scores from the ADOS, but we do not have the cutoff scores for each participant at hand and this is beyond the scope of this paper and was not included within the research protocol as a data element we would collect. Additional details were provided in the manuscript.
  4. Move the total participant numbers cited in the results up to the methods section. – While we recognize that there are differences of opinion about this, generally, the number of participants is a result, not a component of the methods.

Results:

  1. Line 163, “Among the ten VR session completers, seven participants (70%) had mild ASD, two participants (20%) had a moderate level of ASD, and one participant (10%) had a severe level of ASD. The average age of all enrolled participants was 12.53 (SD 2.24) years old.” How was this justified and defined based on endophenotype? This comment was difficult for us to understand what was being asked. We think this was regarding the ASD classification of severity. This was added to the methods to clarify: “The classification of ASD severity was determined clinically, using the Autism Diagnostic Observation Schedule (ADOS),(10) administered in our clinic by trained professionals, after a thorough interview by both a psychologist and a developmental-behavioral pediatrician. The cut-off scores from the ADOS were used to determine ASD severity.”

Discussion:

  1. It is well established that individuals with ASD pay more attention to the eyes than the mouth----this is not surprising----the authors need to tie in their findings more closely with existing literature----what is new that this approach adds to our current knowledge? Cite the relevant literature and connect in terms of where the use of video and visually based interventions are headed. The discussion was substantially revised to address these concerns. Additional references were added as suggested.

Reviewer 2 Report

The studies on VR applications in the ASD field are snowballing. This study tries to understand gaze fixation and visual searching as a VR training approach in children with ASD.

The topic is interesting, and I suggest the following items be considered in the manuscript.

Line 28: provide more information to clarify what "To be sure" stands for at the beginning of the sentence.

At the end of the introduction, part presents the aim and objections of your study. The present format only indicates a lack of available studies on the topic.

If this is a pilot study, as you mentioned at the beginning of the methodology, saying it in the title will be very helpful.

Line 67: By using conducted in the space of one year, do you mean in the round of one year?

Line 73: I did not find a source to support a six years old child with ASD as a youth. I suggest youth be substituted with a more general word such as "individuals."

I suggest replacing DSM V with DSM5 because the Diagnostic and Statistical Manual of Mental Disorders, Fifth Edition (DSM-5), adopted an Arabic numeral style instead of its previous Roman style in the 2013 update to the diagnostic and statistical manual of mental disorders.

I also think it is beneficial to clarify why, when the participants had already had a diagnosis of ASD in a clinical setting, the authors used DSM5 criteria once again. It is also very vital to indicate that to diagnose ASD application of a valid diagnostic scale is strongly recommended, and it is doubtful to use the DSM5 criteria solely for diagnosis and determining the level of the severity of the symptoms.

Throughout the information presented in the methodology and participants section, I could not figure out how many individuals were approached and finally recruited! This information needs to be presented in this section, not in the results.

Providing information about the participants' level of verbal communication is helpful.

I suggest reorganizing the presentation in the formation of the discussion part based on the aim and objectives you need to add in the final part of the introduction section.

What is your justification for the difference between the mild and moderate ASD group concerning looking in the eye of the other person? As you indicated, it is against our present knowledge. Do you think you need a more reliable scale for categorizing individuals with ASD based on the symptoms' severity level? Currently, you only used SDM5 criteria for classifying your group. This need to be mentioned in the limitation part.

Author Response

Reviewer 2: The studies on VR applications in the ASD field are snowballing. This study tries to understand gaze fixation and visual searching as a VR training approach in children with ASD. The topic is interesting, and I suggest the following items be considered in the manuscript.

Line 28: provide more information to clarify what "To be sure" stands for at the beginning of the sentence. – This was modified to state, “Despite the rising prevalence of ASD, . . . “

At the end of the introduction, part presents the aim and objections of your study. The present format only indicates a lack of available studies on the topic. – We appreciate this suggestion. The last part of the introduction was modified to, “The objective of this study was to describe gaze fixation characteristics and visual searching behaviors within a sample of children/youth with ASD. First, we wanted to describe these visual outcomes in respect to a virtual avatar, as well as other locations in a virtual room. Secondarily, we wanted to describe the visual outcomes based on ASD severity. De-scribing these characteristics and behaviors are important as gaze and visual searching could be valuable targets for social skills training and provide objective metrics of change in response to intervention.”

If this is a pilot study, as you mentioned at the beginning of the methodology, saying it in the title will be very helpful. – this was added to the title

Line 67: By using conducted in the space of one year, do you mean in the round of one year? – The study was completed in 12 months. The language was clarified in the manuscript.

Line 73: I did not find a source to support a six years old child with ASD as a youth. I suggest youth be substituted with a more general word such as "individuals." – this was modified to read,  “children and youth.”

I suggest replacing DSM V with DSM5 because the Diagnostic and Statistical Manual of Mental Disorders, Fifth Edition (DSM-5), adopted an Arabic numeral style instead of its previous Roman style in the 2013 update to the diagnostic and statistical manual of mental disorders. – thank you. This was corrected.

I also think it is beneficial to clarify why, when the participants had already had a diagnosis of ASD in a clinical setting, the authors used DSM5 criteria once again. It is also very vital to indicate that to diagnose ASD application of a valid diagnostic scale is strongly recommended, and it is doubtful to use the DSM5 criteria solely for diagnosis and determining the level of the severity of the symptoms. Thank you for this suggestion. The following was added to the methods for clarification: “The classification of ASD severity was determined clinically, using the Autism Diagnostic Observation Schedule (ADOS),(10) administered in our clinic by trained professionals, after a thorough interview by both a psychologist and a developmental-behavioral pediatrician. The cut-off scores from the ADOS were used to determine ASD severity.”

Throughout the information presented in the methodology and participants section, I could not figure out how many individuals were approached and finally recruited! This information needs to be presented in this section, not in the results. – we recognize that there are differences of opinion on this, but in general, the numbers of participants are a result of the method, not a method.

Providing information about the participants' level of verbal communication is helpful. This is beyond the scope of this work and was not collected as a component of the research study. The additional details regarding how severity classification was made (provided above) should provide a more detailed picture of the clinical presentation of the participants.

I suggest reorganizing the presentation in the formation of the discussion part based on the aim and objectives you need to add in the final part of the introduction section. – The introduction, as modified above, provides more clarification throughout the presentation of the results and discussion, adding to the logical flow of the paper. Additional sections were added within the discussion to follow the same flow as the rest of the paper. Additional references were added in the discussion to bolster the paper as recommended.

What is your justification for the difference between the mild and moderate ASD group concerning looking in the eye of the other person? As you indicated, it is against our present knowledge. Do you think you need a more reliable scale for categorizing individuals with ASD based on the symptoms' severity level? Currently, you only used SDM5 criteria for classifying your group. This need to be mentioned in the limitation part. The diagnostic classification details were provided in the methods. We suggest that the VR environment may not present a similar experience to individuals with ASD. This paragraph was modified in the discussion as follows, “It may be that the VR environment was so novel that participants with mild ASD were enthralled with the VR environment and visually searched the room to “take it all in,” rather than interacting with the avatar as we had hoped. It was, however, confirmatory, that participants with moderate ASD displayed a tendency to shift their gaze to the floor as a preferential location, a behavior not observed in the participants with mild ASD.”

Round 2

Reviewer 2 Report

The presently submitted version is an improved version of the study. Most of my comments are addressed and I think that the paper in its present format echoes the ideas and aims of the authors in a more clear format. 

Regarding my past comment about the sample size that is not presented in the sampling section of the study but in the findings. The authors insisted on their approach without presenting any sources for their format of presenting the final sample size in the findings. 

I still think that the final sample size of the study needs to be presented before findings and analysis and for my argument, I can present different sources. 

https://www.jmu.edu/uwc/files/link-library/empirical/findings-results_section_overview.pdf

https://hal.archives-ouvertes.fr/hal-02546796/document

The Results or Findings section in an empirical research paper describes what the researcher(s) found when they analyzed their data. The sample and sampling approach need to be presented in the methodology section under the sample subheading and then analysis and findings are done based on the recruited and defined sample. The findings section's primary purpose is to use the data collected to answer the research question(s) posed in the introduction. For your information, I have presented my comments and your answer in the following.

Throughout the information presented in the methodology and participants section, I could not figure out how many individuals were approached and finally recruited! This information needs to be presented in this section, not in the results. – thank you for this suggestion, we recognize that there are differences of opinion on this, but in general, the numbers of participants are a result of the method, not a method.

I still recommend to bring the first three sentences of the Results in which you have described your final sample to 2.3. subsection. You can, of course, present statistical information such as mean median or SD in the results section the way it is now, but readers need to know at the sampling approach what is the final number of the sample that you have recruited through your explained method and inclusion and exclusion criteria.

Author Response

The presently submitted version is an improved version of the study. Most of my comments are addressed and I think that the paper in its present format echoes the ideas and aims of the authors in a more clear format. 

Thank you.

Regarding my past comment about the sample size that is not presented in the sampling section of the study but in the findings. The authors insisted on their approach without presenting any sources for their format of presenting the final sample size in the findings. 

I still think that the final sample size of the study needs to be presented before findings and analysis and for my argument, I can present different sources. 

https://www.jmu.edu/uwc/files/link-library/empirical/findings-results_section_overview.pdf

https://hal.archives-ouvertes.fr/hal-02546796/document

The Results or Findings section in an empirical research paper describes what the researcher(s) found when they analyzed their data. The sample and sampling approach need to be presented in the methodology section under the sample subheading and then analysis and findings are done based on the recruited and defined sample. The findings section's primary purpose is to use the data collected to answer the research question(s) posed in the introduction. For your information, I have presented my comments and your answer in the following.

Throughout the information presented in the methodology and participants section, I could not figure out how many individuals were approached and finally recruited! This information needs to be presented in this section, not in the results. – thank you for this suggestion, we recognize that there are differences of opinion on this, but in general, the numbers of participants are a result of the method, not a method.

I still recommend to bring the first three sentences of the Results in which you have described your final sample to 2.3. subsection. You can, of course, present statistical information such as mean median or SD in the results section the way it is now, but readers need to know at the sampling approach what is the final number of the sample that you have recruited through your explained method and inclusion and exclusion criteria.

This suggestion was taken and the paper was modified as suggested.